# Should Pharma Companies Waive Their COVID-19 Vaccine Patents? A Legal and Ethical Appraisal

**Tammy Cowart** [1],* , **Tsuriel Rashi** [2],* and **Gregory L. Bock** [3],*

1 Department of Accounting, Finance & Business Law, Soules College of Business, The University of Texas at Tyler, Tyler, TX 75799, USA
2 School of Communication, Ariel University, Ariel 40700, Israel
3 Department of Literature and Languages, College of Arts and Sciences, The University of Texas at Tyler, Tyler, TX 75799, USA
* Correspondence: tcowart@uttyler.edu (T.C.); tsurielra@ariel.ac.il (T.R.); gbock@uttyler.edu (G.L.B.)

**Abstract:** Pharmaceutical companies, like many other types of companies, are incentivized to create, manufacture, and distribute new products, in part due to the legal protections of patent law. However, the tension between patent rights and the public good has been heightened as pharma companies developed new vaccines to combat the COVID-19 pandemic. Wealthy governments paid well for vaccines and received ample supplies, while low- and middle-income countries struggled to obtain access to any vaccines. Some countries called for pharmaceutical companies to waive their patent protections for vaccines in order to facilitate the worldwide manufacture and distribution of COVID-19 vaccines. This paper will examine the rationale of patent protection and patent waiver issues, then compare these concepts with ethical constructs and a Jewish perspective.

**Keywords:** COVID-19 vaccine; patent waiver; Jewish ethics; duty theory





## 1. Introduction of COVID-19 Vaccine Issues

One of Albert Einstein's remarks is engraved near his statue in front of the National Academy of Sciences in Washington: "The right to search for truth implies also a duty; one must not conceal any part of what one has recognized to be true." This sentence highlights two opposing approaches to scientific discovery: the concealment approach, which recognizes the right to secrecy in respect of acquired knowledge, and the disclosure approach, which holds that all acquired knowledge is a public resource and that there is a moral obligation to disclose it for the advancement of mankind.

Patent rights allow a manufacturer or seller of a product to secure exclusive rights to produce and sell a patented product for a period of time in order to recoup costs and profit from the developed property rights. Perhaps no industry relies more heavily on patent rights than the pharmaceutical industry. Research and development, testing, and regulatory approval results in high fixed, sunk costs to the company. Some companies argue the need to recoup these costs through exclusively selling or licensing their patented drugs. Davey writes, "The pharmaceutical industry often cites a need to recoup high research and development (R&D) costs", even though the industry enjoys substantial profits (Davey 2022) Patent protections and government funding have provided companies the motivation to produce COVID vaccines in record time (Melimopoulos 2022). In the United States, the U.S. Patent and Trademark Office launched the COVID-19 Prioritized Examination Pilot Program in May of 2020. It waived payment of certain fees and expedited applications for priority examination for up to 500 patent applications related to COVID-19. "Large scale outbreaks temporarily heighten the incentives for private pharmaceutical companies to engage in costly R&D. These incentives, such as increased funding streams, ignite a race for the development of a vaccine that responds to the outbreak" (Johnson 2022).



In addition to patent prioritization programs, there have been Product Development Partnerships and Technology Access Pools with the World Health Organization and Medicine Patent Pools backed by the United Nations. Organizations like COVAX and Access to COVID-19 Tools Accelerator (ACT) have entered into vaccine supply agreements with countries and drug companies, with mixed results (Chang 2023).

However, poorer countries have had less access to the COVID-19 vaccine. While six billion vaccine doses were preordered by higher-income countries, the World Health Organization reports that less than 1% of the produced vaccines were distributed to developing countries (Aquino 2022). As of May 2021, only about 2% of Africa's 1.2 billion people have been vaccinated (A Patent Waiver on COVID Vaccines is Right and Fair 2021). Africa also imports 99% of its vaccines, which exacerbates this problem. With a tremendous increase in vaccine campaigns in the summer of 2022, 282 million people in Africa received the first COVID-19 vaccine dose, representing 21% of the population of Africa (WHO Africa 2022). However, for comparison, almost 80% of the U.S. population has received one dose of the vaccine (Centers for Disease Control and Prevention 2022), and 72% of Israel's population has received one dose (Mathieu et al. 2021). As of May 2023, the percentage of Africa's population with one vaccine dose has only improved to 36.69%, compared with 75.13% of the European Union, 71% of Israel, and 81% of the United States (Mathieu et al. 2021).

In October 2020, India and South Africa presented a proposal to the World Trade Organization (WTO) for its member nations to waive patent rights to vaccines so that more people could have access. Arguments for such a waiver are that vaccine research, development, and manufacturing are concentrated in a small group of high- and middle-income countries. Companies in these countries are the main patent holders and have sold the majority of their available vaccine doses to their own governments. A vaccine patent waiver would allow lower-income countries to develop their own vaccines without concern over patent infringement lawsuits. In early 2021, the United States, Russia, and China announced support of the patent waiver proposal.

However, other governments like Japan, South Korea, the United Kingdom, and the European Union members have thus far opposed patent waivers for COVID-19 vaccines. Opponents argue that patent waivers provide a shortcut for competitors to acquire access to expensive development technology. They also argue that waivers will not solve the problem, since manufacturing materials are in short supply and developing the manufacturing capability to produce vaccines can take years to develop, including the need to pass clinical trials (Ouellette 2021). These countries also argue that the United States has blocked exports of COVID-19 vaccines to other countries, which has created a distribution problem, as opposed to a vaccine manufacturing problem. However, apparently there are some smaller production facilities that could produce the vaccines quickly if the patent were waived (Silk 2021). In June 2022, the WTO did reach a more limited agreement to ease protections for COVID vaccines. The Ministerial Decision on the TRIPS Agreement provides a specific, partial waiver of patent rights on vaccines and allows Member countries to "use the subject matter of a patent required for the production and supply of COVID-19 vaccines without the consent" of the patent holder. The WTO Decision will apply for five years and will be reviewed annually (WTO 2022). However, experts report that the agreement does not address the knowledge required for manufacturers to produce vaccines in lower-income countries (Robbins 2022), and the language which limits the waiver to the COVID-19 pandemic addresses pharmaceutical companies' concerns that the waiver could be adopted to use mRNA technology to develop vaccines in other fields (Correa and Syam 2022). Moreover, the WTO Ministerial Decision delayed ruling on whether the TRIPS waiver of patent rights would apply to COVID-19 diagnostics and therapeutics. The impact of patent limitations on therapeutics available to developing countries could be more significant than it has been on vaccines (WTO 2023). In March 2023, the WTO again postponed a decision on diagnostic and therapeutic patents, while some members continued to argue that access to vaccines alone are an insufficient solution to the COVID-19 problem (WTO 2023).

The issue of vaccine patent waivers is extremely critical because vaccination has become a medical, social, legal, moral, and religious obligation in many countries around the world (Rashi 2021b). This obligation means that it is very difficult to continue the routine of life without the vaccine against COVID-19, and the creation of the high—and perhaps even complete—dependence on the manufacturers of the vaccine (Rashi 2021a). The registration of the patent by the patent manufacturers gives them tremendous power vis-à-vis entire countries and continents, and on the other hand, the nonregistration of the patent may have a chilling effect in the future for private manufacturers and pharmacology companies who are considering investing in the market of drug and vaccine development and are afraid of the delay by the various regulators around the world. As one economic study noted, intellectual property right enforcement encourages foreign companies to market their products in developing countries, "but it brings with it the potential of substantial price increases of patented products". Conversely, "price regulation can prevent patent holders from exploiting their market power, but not without diminishing the incentives for such firms to expand their operations in the developing world" (Chaudhuri et al. 2006).

Does this demonstrate a flawed view of global health, with vaccines being treated as a market commodity rather than a public good? Others would argue that innovation will be stifled without patent incentives. With over six million deaths from COVID-19 worldwide, we argue that the moral principle of beneficence is of greatest importance in deciding to issue patent waivers. However, we will also consider other principles, such as a justice approach and the right to intellectual property protections. We examine these principles in the context of the benefits and costs of patent protections and waivers, and how patent protections and waivers comport with Jewish ethical principles. One of the perceived lacunae in modern law and ethics is the lack of a comprehensive ethical framework that integrates moral principles and values with religious or spiritual traditions. Some scholars argue that modern secular ethics often fail to provide an adequate foundation for ethical decision-making, as they are based on individualistic, relativistic, or utilitarian assumptions that may not be universally applicable or morally compelling.

Jewish ethics, on the other hand, offers a rich tradition of moral teachings and practices that emphasize the importance of ethical relationships and communal responsibility. Jewish ethics also recognizes the complexity and ambiguity of ethical dilemmas and provides tools for ethical reasoning and moral discernment (Dorff 2003). The relevance of Jewish ethics extends beyond the Jewish community. While Jewish ethics may have originated within a specific religious worldview, its principles and values can be relevant and meaningful for all people, regardless of their beliefs or background.

## 2. Legal Overview of Patent Protections and Waivers

The United States Constitution gives Congress the power to enact patent laws "to promote the progress of science and useful arts, by securing for limited times to authors and inventors the exclusive right to their respective writings and discoveries" (U.S. Const. art. 1). The Patent Act was enacted in 1790, generally revised in 1952, and most recently revised in the Leahy-Smith America Invents Act of 2011. Issuance of a patent allows the patent holder to "exclude others from making, using, offering for sale, or selling" the patented invention (Leahy-Smith America Invents Act 2011).

The United States Patent and Trademark Office (USPTO), part of the Department of Commerce, has the power to issue patents on behalf of the United States Government. Applications for patents are filed with the USPTO, where they are examined to determine if the patents should be issued. To receive a patent, the applicant must show that the invention is new, useful, and nonobvious. Drug patents grant the right of exclusivity for a period of twenty years from the date the application is filed; however, because a substantial portion of the patent period may be consumed with clinical trials and FDA reviews, the patent period may be extended for an additional five years (Beall et al. 2019). Drug makers attempt to recoup costs during the patent period of exclusivity, before the patent period expires, allowing competition from other entities and generic drug competitors which

drives down prices. It is worth noting that some companies engage in "evergreening", the process of tweaking molecules in a drug's formula to extend the life of the patent. Another tactic has been "pay-for-delay" negotiations to keep generic drugs off the market (Davey 2022).

There are ways that patents can be utilized by nonowners. One way is a voluntary license. A voluntary license is a contractual agreement between the patent holder and a third party. In the case of vaccines, this would involve a contractual license between the patent holder and manufacturers for the purpose of developing a generic version of the medicine. The license arrangement typically involves the transfer of technology or knowledge required to produce the vaccine or drug according to the royalty or premium decided in the license agreement. Note that the license can be exclusive to the patent licensee or nonexclusive, allowing the patent holder to license to multiple licensees (Sparsh 2021). While this approach works well in conventional situations, the onus to negotiate with licensees is based on the willingness of the patent owner to do so, which does not bode well for the low-and middle-income countries involved in negotiating for COVID-19 vaccines from vaccine patent owners.

The second type of license is a compulsory license, which occurs when a government grants a right to exploit a patent without the permission of the patent owner (Bird 2009). When this occurs, governments authorize a company to manufacture, use, or distribute the patented invention without the owner's consent. While the United States has codified compulsory licenses and provides reasonable compensation for lost profits and reasonable royalties (Mitchell 2007), the most relevant guidance for the global COVID pandemic is the World Trade Organization's (WTO) Doha Declaration. The WTO has a membership of 164 countries and members agree to intellectual property rules as part of a multilateral trading system, which includes guidelines for patent protection in member countries. In 2001, the WTO amended the Agreement on Trade Related Aspects of Intellectual Property Rights (TRIPS) Agreement to "promote access to medicines once and for all" (WTO 2001) in imposing guidelines and restriction for compulsory licenses for intellectual property. The compulsory licensing provisions of the Doha Declaration have had both positive and negative effects on low- and middle-income countries (Padmanabhan 2021). While some countries have successfully reduced their cost for important drugs or produced generic drugs using compulsory licenses, others have found that pharmaceutical companies are more reluctant to engage with countries that readily use compulsory licenses. Critics of compulsory licensing argue that it deters pharmaceutical research and development and allows government-sanctioned infringement of intellectual property.

A license, either voluntary or compulsory, may be a solution for countries wishing to develop a COVID vaccine, but the temporary waiver of TRIPS patent protections may be the most effective way to loosen "patent protections without disparaging the rights of millions of people across the globe" (Johnson 2022). As stated earlier, the world's vaccine-producing countries are currently divided on the issue of whether there should be a patent waiver for COVID-19 vaccines. Aside from the legal issues discussed herein involving TRIPS and WTO provisions, there are ethical questions about whether vaccine producers have an obligation to waive their patents during this pandemic. The following sections will address the ethical issues involved in vaccine patent waivers.

## 3. Jewish Ethics Regarding Intellectual Property

According to Jewish tradition, the Ten Commandments were given to Moses at Mount Sinai some 3000 years ago in the presence of the entire Israelite nation (600,000 adult males). The complete Torah, written by Moses toward the end of his life, includes 613 commandments. Around these commandments and accompanying elaborations and clarifications (Written Law) there evolved an Oral Law comprising rabbinic discussions and arguments over the ensuing centuries that ultimately coalesced into the halakhah. In the second century of the Common Era, the period in which the first major codification of Jewish Law, the Mishnah, was written, rabbis set up a major center of scholarly religious learning

to facilitate continuation of the halakhic tradition. In the ensuing eighteen centuries, generations of religious leaders living in many countries around the world under the influence of various religiously oriented civilizations (primarily Catholic, Eastern Orthodox, Moslem, and Protestant) continued to broaden and further clarify and codify the Halakhah, which is a praxis-based code of law (i.e., legal principles are derived from specific problems and issues that arise in daily life, much like English Common Law). Thus, during the past 1500 years, tens of thousands of common-man "questions" and local rabbinical "answers" (in Hebrew: "shut") have "clarified" the halakhah, thereby developing what has come to be called "Responsa literature". Every so often, owing to the unwieldiness of such a large corpus, major rabbinical commentators have taken it upon themselves to "codify" the law in some systematic and quasi-authoritative fashion (Rashi 2012).

In his monumental work on the laws of property, the first Chief Rabbi of the State of Israel, Rabbi Yitzhak Isaac HaLevi Herzog (1888–1959), emphasized that the approach of Jewish law—from the very beginning to the present day—has been the development of the law based on concrete examples. Thus, it is difficult to find a clear definition of property rights in general and certainly not of intellectual property rights anywhere in the Written Torah, by way of the Oral Law, the Mishnah, the Talmud, and the Responsa literature. Despite this, Rabbi Herzog insisted that it was possible to find in the sources the intellectual core to provide guidance, even in general terms, for the right religious determination concerning intellectual property:

> It should be made clear once more, that even the sources adduced above would furnish no firm legal ground for patent rights. They would merely supply the spirit and the trend, which would have to be clothed with the body and substance of takkanoth, of legislative enactments . . . Had disputes about such matters been of relatively frequent occurrence, they would have found an echo in our juristic literature, and although, as already noted, there is no direct ruling or dictum in the Talmudim on patent rights, there is in that ocean of Jewish law and lore enough of the basic moral idea and even of a legalistic nucleus to have supplied the authorities with material for dealing with the question from the halakhic standpoint. I have no doubt that under Jewish law had the question become actual, patent rights would have been protected in some measure, at least by special enactments supported by certain Talmudic analogies. (Herzog 1939)

Rabbi Herzog believed it was possible to find an anchor for the rights in a new invention—in fact a patent—from the Babylonian Talmud and its commentaries. Further, it is also possible to base this theoretically without examining specific criteria. At the beginning of his commentary on the Treatise Baba Kama in the Babylonian Talmud—which deals with damages—Rabbi Shimon Shkop (1860–1939; the Rosh Yeshiva of Grodno in Lithuania) referred to one of the primary categories of damage in the Talmud, a pit or hole. He tried to answer the question of why it was necessary to ascribe guilt to a person who dug a pit in the public domain for the damage to someone who is harmed by it. At first glance, it appears that the person did not cause damage directly, since the pit was in the public domain, but according to Rabbi Shkop:

> His pit came about through digging, that is, that he prepared the cause of the damage and for this reason is deemed its owner, like things affecting a person's acknowledged right according to the laws of the Torah and the laws of the world that whoever creates something new in the world is the owner of it for every right, Thus the Torah calls a person who prepares a fault or mishap the owner of a pit and the owner of the fire and is responsible for the damage as the one causing the damage. (Novellae of Rabbi Shimon Shkop, Baba Kama Section 1)

According to Rabbi Shkop, from this reasoning leads to the principle that "someone who invents something new in the world becomes its owner." Rabbi Shkop returned to this principle elsewhere in the context of a discussion about the possibility of acquiring ownership through thought and invention:

Also something that comes about through the wisdom of man has a right to ownership ... since he also acquired it through the tool of thought, the right whereby it is invented by him .... Accordingly since his wisdom recognized it, that is to say, his body and limbs that brought this thing into the world—then immediately when this thing comes into the world it belongs to the owner." (Novellae of Rabbi Shimon Shkop, Gittin, Section 4)

Rabbi Asher Weiss, considered one of the most important rabbis in today's Ultra-Orthodox world, also bases his understanding of intellectual ownership rights and legitimate ownership of a registered patent on logic:

In my humble opinion it appears that it is simple logic that a person also has the monetary rights to the fruits of his brain and his creation, and what is born is not less than what is bought. And just as the fruits of the palm tree belong to the owner of the palm tree, even though he had not purchased them by way of acquisition, so the fruits of his brain are his and no acquisition is required.

Further, in law he ruled that:

[I]n the case of a registered patent, it is clear that it is forbidden by the laws of the Torah for others to work with his format. (Rabbi Asher Weiss, Darkei Hora'a D, 100 (Weiss 2006))

However, alongside recognition of the right of ownership, or quasiownership, that a person has in his invention, we find in Jewish ethics echoes that this right is not unlimited, and that it must be balanced against the public interest, which demands that knowledge resources will be available to all. In the treatise Yoma, the Mishnah provides a list of artisans who did not agree to reveal the secret of their crafts to others, which it censures:

And these are a disgrace: the House of Garmo, which did not want to teach how to prepare the shew breads; the House of Avtinas, which did not want to teach how to prepare the incense; Hugras ben Levi knew the chapter of songs and did not wish to teach it; Ben Kamtzar did not wish to teach how to write ... of these it is said "and the name of the evil ones will rot." (Mishnah, Yoma 3, 11)

The Babylonian Talmud explains that the Mishnah is dealing with professionals, those with specialized knowledge in the workings of the Temple in Jerusalem who refused to disclose their professional secrets: the Garmo family knew how to bake special bread that would remain fresh for a long time; the Avtinas family knew how to light the incense in a special way; Hugras ben Levi knew how to sing in a special way; Ben Kamtzar knew a special calligraphy. The requirement of the religious leadership at that time was that craftsmen should disclose their secrets because, "Everything that God created was created for His own honor. As it is written: 'Everything has been created in My name and in My honor'" (Yoma 38a). This religious approach emphasizes the fact that natural resources are universal because they were created by God. Therefore, it is not right to attach them exclusively to a particular party. This applies to all physical resources, and all the more so to a resource that has medical and curative potential. As related in the Babylonian Talmud in the treatise Avoda Zara:

Rabbi Yoḥanan suffered from the illness tzafdina, which affects the teeth and gums. He went to a certain gentile matron who was a well-known healer. She prepared a medicine for him on Thursday and Friday. Rabbi Yoḥanan said to her: What shall I do tomorrow, on Shabbat, when I cannot come to collect the medicine from you? She said to him: You will not need it. Rabbi Yoḥanan asked her: If I do need it, what shall I do? She said to him: Take an oath to me that you will not reveal the remedy, and I will tell you so that you can prepare it yourself should you need it. Rabbi Yoḥanan took an oath to her: To the God of the Jews, I will not reveal it. She revealed the remedy to him. On the following day Rabbi Yoḥanan went out and taught it publicly, revealing the secret of the remedy.

On first reading, the story seems to reflect the actions of Rabbi Yochanan in a negative light. Even though he swore to the woman that he would not disclose her professional secret, he broke his promise and revealed her pharmacological secret to the public. However, the version of the story in the Jerusalem Talmud (Shabbat 76b) is more obscure as it adds a surprising postscript concerning the fate of the woman. According to one opinion, she committed suicide, but according to a second opinion, she converted [to Judaism]. Such contradictory results for the same story are rare. What do they mean?

The Talmudic disease of scurvy is described as being fatal. According to most of the Talmudic opinions, the woman's "remedy" was no more than a mixture of barley, olive oil, and salt. Thus, people who were dying from a fatal disease could be saved with the simplest of means, available in their own homes. Thus, this woman acted out of greed and a lack of concern for the deaths that could have been easily prevented. Rabbi Yochanan felt that it was inappropriate that information about the composition of such a simple and available food additive, which could save lives, should be held by the "owner of the secret" to be sold to whomever could afford it. So, he used guile to learn her secret so that he could make it known to the public for free.

According to the first opinion in the Jerusalem Talmud, the woman eventually understood why Rabbi Yochanan went to such embarrassing lengths to learn her secret. She concluded that the Rabbi was right to sacrifice his name on the altar of social solidarity because he knew that there was no other way to learn her secret. On this scale of values, preferring the good of the many over that of the individual, the woman understood that Jewish ethics expresses a lofty morality that should and did take precedence over her inappropriate behavior, and this recognition led her to convert.

According to the second opinion, by committing suicide, the woman was protesting against Rabbi Yochanan's ethical approach, which favored the public interest over an individual's rights to his property. The two results, suicide or conversion, reflect different positions on the question of justifying expropriation of intellectual property for the sake of the public interest. The Jerusalem Talmud, which exceptionally offers two polar opposite results regarding a single event, expresses the position of an unresolved ethical dilemma.

## 4. The COVID-19 Vaccine Issue

In March of 2022, Moderna announced that it expected higher-income countries to respect its intellectual property rights and would license its vaccine patents on commercially reasonable terms (Loftus 2022). Moderna's Stephane Bancel said, "If people have used, or are using our technology to make a vaccine, I don't understand why, once we're in an endemic setting when there's plenty of vaccine and there's no issue to supply vaccines, why we should not get rewarded for the things we invented" (Loftus 2022). However, Moderna has pledged that it would not enforce its patents against the 92 countries that are part of the international Covax program or efforts by Afrigen Biologics and Vaccines, which is working to copy Moderna's vaccine for use in Africa. It is worth noting that patent waivers are not the biggest obstacle for Africa. Even with the patent information, the technology process for vaccine production must be transferred, which takes months. "Merely giving South Africa the instructions to make a shot would be a little like giving a recipe for a molecular-cuisine dish to someone who has only ever made toast—and then expecting them to cook it to a Michelin three-star standard" (Sulcas and Malan 2021). Notably, World Health Organization official Dr. Martin Friede said that Moderna has been reluctant to meet to discuss a license for the patent (Prasad 2021). Moderna received $1 billion from the U.S. government through Operation Warp Speed to develop and produce its COVID-19 vaccine with help from the National Institute of Health (NIH). Yet, Moderna has not listed NIH researchers as coinventors on its COVID vaccine patents, "limiting the influence the government can have over the biotechnology company on matters relating to manufacturing and distribution of the vaccine" (Davey 2022). Pfizer did not accept funding from the U.S. government for development of its vaccine, but did agree to sell doses at a not-for-profit price for donation to low- and middle income countries (Prasad 2021). "When

you get money from someone, that always comes with strings," said Pfizer CEO Albert Bourla (Biden's Moderna Vaccine Double-Cross 2021). However, there is more to the story, as Pfizer's partner BioNTech did accept payment from the German government. Both companies recently announced plans to offer the newest version of the COVID-19 vaccine for about $130 per dose, about four to five times greater than the average price paid by the federal government (Kates et al. 2023).

Moderna is a relatively new company, starting up in 2010, and pioneered mRNA, the technology used for its COVID vaccine. Before 2020, Moderna had invested more than $2.5 billion to develop its vaccine platform but had never really made a profit. It did not have enough cash to conduct the clinical trials for the COVID-19 vaccine, so the U.S. government provided the nearly $1 billion in funds so it could conduct the Phase 1 trial. Nonetheless, some argue that private companies should not be allowed to receive taxpayer support to develop technologies that are later restricted from the public (Kapczynski 2021).

Despite this discussion of patent waivers, some patent disputes have already erupted. Moderna has disputed the NIH's effort to include its scientists as coinventors on the patent application (Loftus 2022). Drug makers Arbutus and Alnylam Pharmaceuticals have also filed suits against Moderna and Pfizer and BioNTech for patent infringement related to the COVID-19 vaccines (Arbutus Files Patent Lawsuit against Pfizer/BioNTech over COVID Vaccines 2023). Finally, despite its pledge that it would not enforce its patent, Moderna has filed suit for patent infringement against Pfizer and BioNTech in both U.S. and German courts (Murphy 2023). Moderna claims that Pfizer and BioNTech's COVID-19 vaccines infringe on patents Moderna filed between 2010 and 2016, using technology that was not in use by Pfizer and BioNTech at the time of the COVID outbreak. Notably, Pfizer's COVID vaccine earned over $36 billion in global sales in 2021, with an expected $33 billion in sales in 2022. Moderna recorded $17.6 billion in revenue in 2021, and analysts project more than $21 billion in 2022 (Murphy 2023).

Pharmaceutical companies operate with an expectation of recovering research and development costs and reaping profits from drug development by relying on patent protection. Indeed, without patent protection, pharma companies might be reluctant to formulate and develop new drugs, or only commit to develop drugs that are likely to result in sales to large populations. This rationale would perhaps meet the needs of the greatest number of people, but it is not necessarily how we expect pharma companies to behave in the market. Indeed, even the U.S. government provides financial incentives for drug companies to develop so-called "orphan drugs" which treat rare illnesses affecting less than 200,000 people (Chua et al. 2021). Government policy should be mindful of balancing the needs of the public with the needs of companies to recoup costs and reap at least reasonable profits. This balancing act has become particularly difficult given the limited drug options available during this global pandemic; however, the ethical discussions presented herein can provide some guidance in dealing with these difficult issues.

## 5. The Duty to Provide Access to Vaccines

The global debate surrounding deferring patents for the COVID-19 vaccine illustrates how complex and tangled this dilemma was and is. From the source literature in Jewish ethics, it emerges that the justification and recognition for the defense of patents derive from the rights of creation. In this sense there were those who said that just as the fruits of a man's hands belong to him, so too the invention of his imagination and bringing it into being. According to this approach, creativity—whether of the hands or the mind—accord a status of "ownership" to an invention and its fruits.

Based upon the Talmudic story concerning invention in the field of medicine, Rabbi Herzog emphasized that Jewish ethics does not look kindly on a person's desire to keep his invention to himself, without limitation. This considers that we are dealing with a lifesaving discovery (Herzog 1939). The tension is between the notions that the craftsman is entitled to receive money for his property but should not be permitted to keep the secret of the patent to himself, and certainly not forever.

It is interesting to note that the three halakhic decisors quoted—Rabbi Shkop, Rabbi Herzog, and Rabbi Weiss—wrote between the 19th and 21st centuries, a period in which the field of patents became much more technologically advanced. It is suitable for our discussion to quote the words of an expert in history and in Jewish law and its development, Prof. Edward Fram: "Like rabbis in all periods, . . . rabbis had to address the needs of the marketplace in order to keep the halakhah relevant and maintain the integrity of Jewish life" (Fram 1997).

Thus, the economic question directly affects both manufacturers and customers around the world. The integrity of Jewish ethics points to the theoretical basis not being principally economic, but rather, recognition of man's ability to create, and the connection between himself and his creation while taking into account the needs of society. Hence, there is no doubt that a person is the owner of his creation, but the public interest to protect society's health takes precedence.

Moreover, each one of us is responsible for doing our part. In the famine relief literature, Peter Singer draws an analogy with a child drowning in a shallow pond. If we witness a child drowning and can reach in and save them without "unreasonable cost to oneself," then we have a duty to do so. This argument applies just as much to famine relief as it does to the distribution of vaccines in a pandemic. Each of us should do our small part to help speed up the delivery of the vaccine to the whole world. Singer believes that we ought to go so far as "reducing ourselves to the level of marginal utility" (Wolfe 2015). He has even said that we ought to give away up to a third of our income. If we have an obligation to participate in famine relief, how much more an obligation do we have to help with vaccine distribution in a pandemic, one that continues to destroy lives and devastate communities worldwide?

## 6. Conclusions

Pharmaceutical companies have an obligation to help as well, but waiving patents is not the only morally acceptable solution. The company could take a cut in its profits. The company could increase production, perhaps by using its profits to acquire more production facilities. It could work with local governments and NGOs to provide a lower-cost version of the vaccine, or simply offer a lower-priced version itself for certain countries. Moreover, governments could donate portions of their supply or provide tax incentives for others to do the same. If these options exist, then we can manage the current crisis without patent waivers.

Justice requires world leaders and pharmaceutical companies to address inequities in access to public health resources. This moral burden rests, to some extent, on all of us, but it especially rests on the ones that hold the keys, the ones who created the vaccines. Waiving patents may not be the only solution, but we face a real crisis, and the lack of access to vaccines is a moral emergency. If other options are not successful, and if equitable access is not established, then patents should be waived.

**Author Contributions:** Legal Discussion and analysis: T.C.; Jewish ethics: T.R.; ethical discussion and analysis: G.L.B.; writing and editing: T.C. All authors have read and agreed to the published version of the manuscript.

**Funding:** This research received no external funding.

**Institutional Review Board Statement:** Not applicable.

**Informed Consent Statement:** Not applicable.

**Data Availability Statement:** No new data were created.

**Conflicts of Interest:** The authors declare no conflict of interest.

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
