# Peer review of "Should Pharma Companies Waive Their COVID-19 Vaccine Patents? A Legal and Ethical Appraisal"

_laws_

Round 1
Reviewer 1 Report
This is an interesting paper that uses arguments from Jewish ethics about whether there should be a patent waiver to increase accessibility to COVID-19 vaccines. However, at present there are a number of major problems that need to be addressed.
1. In the Introduction the authors need to explain why they are using ethical arguments that arise out of Jewish teaching. Do Jewish ethics have something special to contribute to this discussion that are not present in other ethical traditions?
2. The Jewish laws of property that the authors cite would seem only to apply to individual ownership. Do they also apply to ownership by corporations?
3. Are the authors just arguing from one stream of Jewish thinking (e.g., orthodox Judaism) or do their arguments come from all of the streams. If the former, then why do the authors believe that the stream that they are arguing from is the "correct" one?
4. The authors are basing their arguments on the assumption that the patent system as it currently exists is the only way to stimulate R&D into new medicines. However, a number of alternatives have been developed, including Product Development Partnerships which is currently being employed by DNDi.
5. The authors have an incorrect understanding of pharmaceutical economics. The money that companies invested in researching and developing new drugs is a sunk cost. Companies are not attempting to recoup these costs when they sell new drugs. The revenue from selling new drugs provides the capital for researching and developing the next generation of drugs.
6. Lines 32-33: COVID vaccines were developed rapidly because of the heavy public investment in R&D and advanced orders from wealthy countries that guaranteed a market, not because of patents.
7. Lines 37-42: Figures about the percent of people in various countries and regions who have been vaccinated that are being used by the authors are 18 months old. They should be able to find more up-to-date figures.
8. Lines 43-47 (and other places): It was not just patent rights for vaccines that were the subject of the proposed WTO waiver but a general waiver of intellectual property rights for technology, manufacturing secrets, etc. Also, what was being proposed was only a temporary waiver until the pandemic was over.
9. Lines 62-64: There are more robust analyses of the limited patent waiver than a New York Times story. See for instance the report from the South Centre - https://www.southcentre.int/wp-content/uploads/2022/11/RP169_The-WTO-TRIPS-Decision-on-COVID-19-Vaccines_EN.pdf.
10. Line 65: am not clear what the authors mean by "delaying the registration of the patent". As they say, what was being proposed at the WTO was a waiver.
11. Lines 72-75: The argument that a waiver available to low- and middle-income countries would discourage investment in new products ignores the fact that the large bulk of the revenue that companies generate comes from sales in high-income countries. Sales in low- and middle-income countries make up a small percent of their revenue.
12. Lines 96-97: The authors should make it clear that patent life is 20 years from the date when the patent application is filed.
13. Lines 98-101: Monopolies on drugs can also be extended through filing multiple patents. For example the original patent on Humira expired in 2014 but the monopoly period has been extended to 2037 because of additional multiple patents.
14. Line 133: Voluntary or compulsory licenses are not sufficient to allow for vaccine manufacturing as they only cover the vaccine itself and not all the other intellectual property needed to be able to produce a vaccine. Moreover, compulsory licenses need to be negotiated with individual countries whereas the need for vaccines spans multiple countries.
15. Lines 275-280: The authors say that patent waivers are not the biggest obstacle for Africa but that ignores the fact that Moderna (and Pfizer) will be substantially raising the price of their vaccine. In addition, while there may not be an acute need for vaccine now as there was in 2021, there will be an ongoing need for vaccines due to the mutation of the corona virus and that means a continual need for large quantities of vaccine that can be met by more distributed production.
16. Lines 287-289: Pfizer did not accept money from the US government, but its partner BioNTech did take money from the German government. Moreover, Pfizer benefited from a substantial advanced market purchase by the US government.
17. Lines 298-323: There is a long discussion about patent disputes in this paragraph that could be considerably shortened. The meat of the argument seems to be on lines 318 to 323.
Author Response
Responses to Reviewer #1
FIRST SET OF RESPONSES:
Reviewer #1 wrote that the authors need to explain why they are using ethical arguments that arise out of Jewish teaching. He wondered if Jewish ethics have something special to contribute to this discussion that are not present in other ethical traditions?
One of the perceived lacunae in modern law and ethics is the lack of a comprehensive ethical framework that integrates moral principles and values with religious or spiritual traditions. Some scholars argue that modern secular ethics often fail to provide an adequate foundation for ethical decision-making, as they are based on individualistic, relativistic, or utilitarian assumptions that may not be universally applicable or morally compelling.
Jewish ethics, on the other hand, offers a rich tradition of moral teachings and practices that emphasize the importance of ethical relationships and communal responsibility. Jewish ethics also recognizes the complexity and ambiguity of ethical dilemmas and provides tools for ethical reasoning and moral discernment.
In the second comment Reviewer #1 mentioned that the Jewish laws of property that the authors cite would seem only to apply to individual ownership. His question was: Do they also apply to ownership by corporations?
indeed. This is an excellent comment, but we tried not to expand the discussion more than necessary. The development of the economy in modern times brought to the world legal entities that were not recognized before. One of them is the corporation.
Even in Jewish law, there was much debate about how to treat this new legal personality of the corporation. Many argued that it is not possible to grant legal status in property law to someone who is not a flesh and blood human being. In their opinion, the corporation is nothing more than an ordinary partnership, and the limitation of the personal liability of its members should not be recognized unless limited by contract. Others have proposed various Talmudic sources (such as Mishna Nadar, Chapter 5, Mishna 4) as a basis for recognizing the separate personality of the corporation.
At the end of the discussion, it emerges from the approach of the Talmud and Hebrew law, then, that besides the separate personality of the corporation, there still remains an element of private partnership. Each of the partners has some degree of ownership in the corporation and it is reflected in several matters. As mentioned, we preferred not to expand the discussion around this issue so as not to divert it from the main issue.
In the third comment Reviewer #1 asked: Are the authors just arguing from one stream of Jewish thinking (e.g., orthodox Judaism) or do their arguments come from all of the streams. If the former, then why do the authors believe that the stream that they are arguing from is the "correct" one?
we are not arguing from one stream of Jewish thinking (e.g., orthodox Judaism) but we only tried to present the opinions of the rabbis and researchers who dealt with the issue of patent protection in terms of ethics and Jewish law. Those who dealt with this issue were mostly from the Orthodox stream. Unfortunately, the discussion on this issue was not significant in Reform or Conservative Judaism.
SECOND SET OF RESPONSES:
4.     Reviewer says: The authors are basing their arguments on the assumption that the patent system as it currently exists is the only way to stimulate R&D into new medicines. However, a number of alternatives have been developed, including Product Development Partnerships which is currently being employed by DNDi.
We have incorporated discussion of PDPs, Technology Access Pools, MPP, COVAX, and ACT.
5.     The authors have an incorrect understanding of pharmaceutical economics. The money that companies invested in researching and developing new drugs is a sunk cost. Companies are not attempting to recoup these costs when they sell new drugs. The revenue from selling new drugs provides the capital for researching and developing the next generation of drugs.
We will incorporate the following quotes into the article. “The pharmaceutical industry often cites a need to recoup high research and development (R&D) costs.” 35 Harv. J. Law & Tech. 689, 690. “The industry still enjoys substantial profit margins and often spends more on advertising and marketing than R&D. Id.
“The need for patents in the pharmaceutical sector has typically been justified by the need to ensure that drug manufacturers are able to recoup the substantial investments necessary for R&D and the costs of regulatory testing.” 53 U. Miami Inter-Am. L.R. 143, 150.
6.     Lines 32-33: COVID vaccines were developed rapidly because of the heavy public investment in R&D and advanced orders from wealthy countries that guaranteed a market, not because of patents.
Will add the following:
“Large scale outbreaks temporarily heighten the incentives for private pharmaceutical companies to engage in costly R&D. These incentives, such as increased funding streams, ignite a race for the development of a vaccine that responds to the outbreak.” 53 U. Miami Inter-Am. L.R. 143, 167.
7.     Lines 37-42: Figures about the percent of people in various countries and regions who have been vaccinated that are being used by the authors are 18 months old. They should be able to find more up-to-date figures.
The disparity in vaccination rates exemplifies the issue that is at the crux of the paper. Nonetheless, we state in the paper that “With a tremendous increase in vaccine campaigns in the summer of 2022, 282 million people in Africa received the first Covid vaccine dose, representing 21% of the population of Africa[4].” This is less than a year old, but we can update it again.
8.     Lines 43-47 (and other places): It was not just patent rights for vaccines that were the subject of the proposed WTO waiver but a general waiver of intellectual property rights for technology, manufacturing secrets, etc. Also, what was being proposed was only a temporary waiver until the pandemic was over.
Language here has been added to include the WTO Decision in June, 2022, Ministerial Decision on the TRIPS Agreement. The Ministerial Decision on the TRIPS Agreement provides a partial waiver of patent rights on vaccines and allows Member countries to “use the subject matter of a patent required for the production and supply of COVID-19 vaccines without the consent” of the patent holder. The WTO Decision will apply for five years and will be reviewed annually. (WTO 22-4786)
9.     Lines 62-64: There are more robust analyses of the limited patent waiver than a New York Times story. See for instance the report from the South Centre - https://www.southcentre.int/wp-content/uploads/2022/11/RP169_The-WTO-TRIPS-Decision-on-COVID-19-Vaccines_EN.pdf.
Thank you for the link. Additional language has been added incorporating this source and WTO:
The Ministerial Decision on the TRIPS Agreement provides a specific, partial waiver of patent rights on vaccines and allows Member countries to “use the subject matter of a patent required for the production and supply of COVID-19 vaccines without the consent” of the patent holder. The WTO Decision will apply for five years and will be reviewed annually. [ ] However experts report that the agreement does not address the knowledge required for manufacturers to produce vaccines in lower income countries[9], and language limited waiver to the COVID-19 pandemic address pharmaceutical companies’ concerns that the waiver could be adopted to use mRNA technology to develop vaccines in other fields[south centre]. Moreover, the WTO Decision postponed a ruling on whether the TRIPS waiver would apply to Covid-19 diagnostics and therapeutics{WTO Decision} The impact of patent limitations on therapeutics could be more significant than it has been on vaccines [south centre], In March 2023, the WTO again postponed a decision on the matter, while some members argued that access to vaccines alone are an insufficient solution to the COVID-19 problem [WTO 2023 news].
10.  Line 65: am not clear what the authors mean by "delaying the registration of the patent". As they say, what was being proposed at the WTO was a waiver.
Wording has been changed.
11.  Lines 72-75: The argument that a waiver available to low- and middle-income countries would discourage investment in new products ignores the fact that the large bulk of the revenue that companies generate comes from sales in high-income countries. Sales in low- and middle-income countries make up a small percent of their revenue.
We agree. We state the following in the paper: Pharmaceutical companies operate with an expectation of recovering research and development costs and reaping profits from drug development by relying on patent protection. Indeed, without patent protection, pharma companies might be reluctant to formulate and develop new drugs, or only commit to develop drugs that are likely to result in sales to large populations.
We also cite an economic study earlier in the paper which found: Conversely, “price regulation can prevent patent holders from exploiting their market power, but not without diminishing the incentives for such firms to expand their operations in the developing world.” [ ]
12.  Lines 96-97: The authors should make it clear that patent life is 20 years from the date when the patent application is filed.
This has been added.
13.  Lines 98-101: Monopolies on drugs can also be extended through filing multiple patents. For example the original patent on Humira expired in 2014 but the monopoly period has been extended to 2037 because of additional multiple patents.
The following language has been added:
It is worth noting that some companies engage in “evergreening,” the process of tweaking molecules in a drug’s formula to extend the life of the patent. Another tactic has been “pay-for-delay” negotiations to keep generic drugs off the market.
14.  Line 133: Voluntary or compulsory licenses are not sufficient to allow for vaccine manufacturing as they only cover the vaccine itself and not all the other intellectual property needed to be able to produce a vaccine. Moreover, compulsory licenses need to be negotiated with individual countries whereas the need for vaccines spans multiple countries.
We agree. We state: “A license, either voluntary or compulsory, may be a solution for countries wishing to develop a Covid vaccine, but the temporary waiver of TRIPS patent protections may be the most effective way to loosen “patent protections without disparaging the rights of millions of people across the globe”[
15.  Lines 275-280: The authors say that patent waivers are not the biggest obstacle for Africa but that ignores the fact that Moderna (and Pfizer) will be substantially raising the price of their vaccine. In addition, while there may not be an acute need for vaccine now as there was in 2021, there will be an ongoing need for vaccines due to the mutation of the corona virus and that means a continual need for large quantities of vaccine that can be met by more distributed production.
Great comment. We have added the following:
Both companies recently announced plans to offer the newest version of the the COVID vaccine for about $130 per dose, about 4 to 5 times greater than the average price paid by the federal government [ ] https://www.kff.org/coronavirus-covid-19/issue-brief/how-much-could-covid-19-vaccines-cost-the-u-s-after-commercialization/.
16.  Lines 287-289: Pfizer did not accept money from the US government, but its partner BioNTech did take money from the German government. Moreover, Pfizer benefited from a substantial advanced market purchase by the US government.
Thank you for this comment. It has been added.
17.  Lines 298-323: There is a long discussion about patent disputes in this paragraph that could be considerably shortened. The meat of the argument seems to be on lines 318 to 323. 
As a litigation attorney, I must advocate that this discussion demonstrates the intent for companies to enforce their patents if they believe that it would benefit them financially. As the pandemic calms, these and even other pharma companies will become bolder in enforcing their patents, as shown by the recent lawsuits we discuss in the paper.
Reviewer 2 Report
The manuscript makes an interesting case for approaching issues regarding patent rights and covid-19 vaccines from the perspective of Jewish ethics, but it should be strengthened along a number of dimesnsions. First, and most crucially, it should somehow be explained why Jewish ethics (which rests on exegesis/interpretation of texts belonging to a specific comprehensive religious worldview) is relevant, if at all, for non-believers or for persons with different religious beliefs. Second, the originality of the argument, in any event, does not appear to be extremely high. We end up with the generally accepted view that the public interest, under certain conditions, trumps IP rights. This seems prosaic (it is accepted almost universally and forms part of the existing legal framework), so it should be better explained how, if at all, Jewish ethics provides a new and interesting perspective to tackle this specific question. If these points were adequately addressed, the paper could later be accepted for publication.
Author Response
Please see the text bellow for a careful reply to the concern about Jewish ethics. Elements of this reply were added to the essay itself to make it more compelling to the audience of Laws. As for the point about a lack of originality in the essay, we believe that addressing the issue from the perspective of Jewish ethics does, in fact, contribute something new to the issue under discussion in that it provides a philosophically and religiously rich account of moral obligations.
Reviewer #2 commented that it should somehow be explained why Jewish ethics (which rests on exegesis/interpretation of texts belonging to a specific comprehensive religious worldview) is relevant, if at all, for non-believers or for persons with different religious beliefs.
We think that the relevance of Jewish ethics extends beyond the Jewish community. Many of the ethical values and principles espoused in Jewish tradition, such as the value of justice, compassion, and respect for human dignity, are universal and can be embraced by people of all faiths or even those who do not subscribe to any particular religious belief.
For example, the Jewish concept of Tikkun Olam, which means "repairing the world," is often invoked to promote social justice. This concept emphasizes the importance of taking action to address the world's problems and to make the world a better place for all people, regardless of their religious beliefs.
Similarly, the Jewish principle of chesed, or loving-kindness, can be seen as a universal value that transcends religious boundaries. Chesed encourages people to show compassion and kindness to others, regardless of their background or beliefs.
In addition, the wisdom and insights contained in Jewish texts and traditions can offer valuable insights into human nature, the nature of the world, and the ethical dilemmas that confront us all.
Overall, while Jewish ethics may have originated within a specific religious worldview, its principles and values can be relevant and meaningful for all people, regardless of their beliefs or background.
A summary of these ideas has been added to the article.
Round 2
Reviewer 1 Report
The revisions that the authors have undertaken have dealt with most of my concerns but there are two points that need to be resolved:
1. I am not sure, despite the addition of the Davey reference, if the authors are agreeing with the statement that companies need to recoup R&D costs through sales of their drugs. As I said, these are sunk costs and are not recouped through sales.
2. Lines 36-38: Much of the technology that was used in the development of the mRNA vaccine was publicly funded especially in the case of the Moderna vaccine. Therefore, the statement that patent protection incentivized the development of vaccines in record time is not completely correct. The existence of patent protection may have led to the application of the existing basic research quickly but that is not the same as the development of a vaccine de novo.
Author Response
1. Added: Research and development, testing, and regulatory approval results in high fixed, sunk costs to the company. Some companies argue the need to recoup these costs through exclusively selling or licensing their patented drugs.
2. Line 35 - "develop" has been deleted
Reviewer 2 Report
The authors did a good job responding to my remarks and I therefore recommend the paper for acceptance and publication.
Author Response
Thank you for your comments.